# The Protective Effect of Chlorogenic Acid on Vascular Senescence via the Nrf2/HO-1 Pathway

**DOI:** 10.3390/ijms21124527

**Published:** 2020-06-25

**Authors:** Yoshiko Hada, Haruhito A. Uchida, Nozomu Otaka, Yasuhiro Onishi, Shugo Okamoto, Mariko Nishiwaki, Rika Takemoto, Hidemi Takeuchi, Jun Wada

**Affiliations:** 1Department of Nephrology, Rheumatology, Endocrinology and Metabolism, Okayama University Graduate School of Medicine, Dentistry, and Pharmaceutical Science, Okayama 700-8558, Japan; yuzuro7@gmail.com (Y.H.); nomu2129@gmail.com (N.O.); oocyst0024@gmail.com (Y.O.); shubo422@gmail.com (S.O.); m.tsuchida.okayama.u@gmail.com (M.N.); rika-t@md.okayama-u.ac.jp (R.T.); takeuchih@okayama-u.ac.jp (H.T.); junwada@okayama-u.ac.jp (J.W.); 2Department of Chronic Kidney Disease and Cardiovascular Disease, Okayama University Graduate School of Medicine, Dentistry, and Pharmaceutical Science, Okayama 700-8558, Japan; 3Department of Human Resource Development of Dialysis Therapy, Okayama University Graduate School of Medicine, Dentistry, and Pharmaceutical Science, Okayama 700-8558, Japan

**Keywords:** chlorogenic acid, vascular senescence, nuclear factor erythroid 2-related factor 2, heme oxygenase-1

## Abstract

The world faces the serious problem of aging. In this study, we aimed to investigate the effect of chlorogenic acid (CGA) on vascular senescence. C57/BL6 female mice that were 14 ± 3 months old were infused with either Angiotensin II (AngII) or saline subcutaneously for two weeks. These mice were administered CGA of 20 or 40 mg/kg/day, or saline via oral gavage. AngII infusion developed vascular senescence, which was confirmed by senescence associated-β-galactosidase (SA-β-gal) staining. CGA administration attenuated vascular senescence in a dose-dependent manner, in association with the increase of Sirtuin 1 (Sirt1) and endothelial nitric oxide synthase (eNOS), and with the decrease of p-Akt, PAI-1, p53, and p21. In an in vitro study, with or without pre-treatment of CGA, Human Umbilical Vein Endothelial Cells (HUVECs) were stimulated with H_2_O_2_ for an hour, then cultured in the absence or presence of 0.5–5.0 μM CGA for the indicated time. Endothelial cell senescence was induced by H_2_O_2_, which was attenuated by CGA treatment. Pre-treatment of CGA increased Nrf2 in HUVECs. After H_2_O_2_ treatment, translocation of Nrf2 into the nucleus and the subsequent increase of Heme Oxygenase-1 (HO-1) were observed earlier in CGA-treated cells. Furthermore, the HO-1 inhibitor canceled the beneficial effect of CGA on vascular senescence in mice. In conclusion, CGA exerts a beneficial effect on vascular senescence, which is at least partly dependent on the Nuclear factor erythroid 2-factor 2 (Nrf2)/HO-1 pathway.

## 1. Introduction

The lifespan of people is increasing worldwide. Since the pace of population ageing around the world is getting faster, the prevalence of age-related diseases is on an upward trend. Thus, aging raises socioeconomic problem for countries around the world. In this sense, our society will face an enormous economic challenge in the decades to come. Consequently, there is an urgent need to find suitable interventions that slow down aging and reduce or delay the incidence of debilitating age-related diseases.

Among senescence-related conditions, vascular senescence has been identified as an important factor underlying diseases such as hypertension, stroke, and arteriosclerosis [1,2,3]. Vascular senescence not only reduces the function of the affected organ but also exacerbates insufficiency due to chronic inflammation [4]. Therefore, aging of the vasculature plays a central role in senescence-related diseases [5,6]. Among the vascular component cells, vascular endothelial cells (ECs) play a pivotal role in maintaining vascular homeostasis and health and respond to physical and chemical stimuli. Thus, therapies targeting endothelial senescence would have important clinical implications for the treatment of senescence-related diseases including cardiovascular diseases [7,8]. Furthermore, three main routes lead to senescence: telomere-dependent replicative senescence, oncogene-induced senescence, and stress-induced (premature) senescence [9]. The mechanisms of senescence are very complex, and thus remain poorly understood. Since many cell types never exhaust their maximum replicative potential during their lifespan and fail to present senescence, premature senescence is likely the most important inducer of cellular senescence in vivo [10]. Although cellular models of senescence provide valuable mechanistic information, they can lead to limited data because they may not replicate in vivo biology. Animal models are useful tools for an investigation on aging, allowing us to find conserved pathways that may be involved in human aging. Activation of the renin-angiotensin system induced by AngII can act as a key player in cell and organ senescence [11,12]. Interestingly, AngII-affected genes are abundant in vascular senescence pathway. [13] Indeed, many papers have reported vascular senescence induced by AngII [14,15,16].

Polyphenols are compounds which are contained in many fruits, vegetables, and beverages including tea or wine [17]. In addition, a few polyphenols are widely accepted to suppress vascular changes associated with aging [18,19,20]. Coffee is a very popular beverage in the world and has been consumed for its attractive flavors and physiological effects. Several studies have reported a relationship between the consumption of coffee and the potential health benefits, which might be correlated with polyphenols [21,22,23]. Recent investigations have demonstrated that coffee showed a 24% of risk reduction in total mortality in subjects who consumed 3–4 cups a day compared in non-drinkers; additionally, the consumption of coffee was inversely associated with mortality from heart disease and cerebrovascular disease [24]. One of the most abundant polyphenols in coffee is chlorogenic acid [25]. Moreover, it has been found that chlorogenic acid has anti-oxidant [26], anti-inflammatory [26], anticancer [27], antidiabetic [28], antihypertensive [29], and antineurodegenerative activities [30,31]. Despite these promising and diverse antisenescence actions, investigations addressing the effect of CGA on senescence are scarce.

Nrf2 was cloned in 1994 by Kan et al. as a factor that binds to the NE-F2 coupling array in the globulin gene expression control region [32]. Recently, Nrf2 has been successively reported to play important roles in detoxification, oxidative stress, and inflammation. Attention is focused on the role of Nrf2 in aging research, since several studies have revealed that Nrf2 attenuates oxidative stress and inflammation, both of which are known to lead to aging. Moreover, Nrf2 has been demonstrated to play a role in slowing aging processes by mediating the beneficial effects of many manipulations that extend longevity and health span [33,34]. Several polyphenols, such as resveratrol and curcumin, have been reported to attenuate aging via the Nrf2/HO-1 pathway [35,36,37]. In addition, CGA has been reported to enhance the Nrf2/HO-1 pathway in several cells [38].

This study is aimed to test the hypothesis that CGA protects against vascular senescence and to explore the role of the Nrf2/HO-1 pathway in vivo and in vitro.

## 2. Results

### 2.1. CGA Inhibits AngII-Induced Vascular Senescence

To investigate the effect of CGA on vascular senescence in vivo, CGA was administered to mice. No significant difference was found in body weight and heart rate among the saline/saline, the saline/CGA, the AngII/saline, and the AngII/CGA groups (Table 1). Systolic blood pressure (SBP) was notably higher in the AngII/saline group compared with the saline/saline and the saline/CGA groups (Table 1).

The aorta was dissected from these mice, and the phenotype of senescence with and without CGA administration was evaluated by SA-β-gal assay. SA-β-gal staining increased in AngII-infused mice compared to saline-infused mice. The mice treated with CGA suppressed AngII-induced senescence in ECs in a dose-dependent manner (Figure 1).

### 2.2. Treatment with CGA Attenuates H_2_O_2_-Induced Cellular Senescence in HUVECs

Next, to examine the preferable effect of CGA in vitro, HUVECs were treated with H_2_O_2_ to induce senescence. H_2_O_2_ increased the number of 8-hydroxy-2′-deoxyguanosine (8-OHdG)-positive cells, suggesting that the DNA damage level increased in HUVECs. Treatment with CGA reduced the number of 8-OHdG-positive cells in a dose-dependent manner (Figure 2a). In addition, H_2_O_2_ induced flattened morphology and increased SA-β-gal activity. Treatment with CGA attenuated SA-β-gal activity and restored the morphological appearance of senescence in a dose-dependent manner (Figure 2b). Furthermore, H_2_O_2_ reduced cell proliferation (Figure 2c). Treatment with CGA at 1.0 µM abrogated the suppression of cell proliferation by H_2_O_2_. However, 5.0 μM CGA led severe DNA damage, flattened morphology, enhanced SA-β-gal activity, and reduced cell proliferation, indicating toxicity. Thus, 0.5 and 1.0 µM concentrations of CGA were used for further experiments. To investigate the effect of CGA without H_2_O_2_, HUVECs were exposed to different concentrations of CGA for three days, and then the Sirt1 and eNOS were assessed. The expression of Sirt1 and eNOS increased in a dose-dependent manner related to CGA (Figure 2d).

### 2.3. CGA Exerts a Favorable Effect on Senescence-Related Markers

Exposure to H_2_O_2_ led to a 40–50% reduction in the expressions of Sirt1 and eNOS in HUVECs. However, co-incubation with CGA significantly increased the expressions of Sirt1 and eNOS compared with the CGA-untreated groups (*p* < 0.05_,_
Figure 3). Exposure to H_2_O_2_ significantly increased the expressions of plasminogen activator inhibitor-1 (PAI-1), p53, and p21. Co-treatment with CGA significantly attenuated their increases (*p* < 0.05, Figure 3).

### 2.4. CGA Induces Nrf2 and HO-1 Expression

To further investigate the anti-senescence mechanism of CGA, HUVECs were exposed to different concentrations of CGA for three days. The expressions of Nrf2 and HO-1 were examined: 1.0 μM CGA significantly increased the protein level of Nrf2 (Figure 4a). However, mRNA levels of Nrf2 and keap1 showed no significant changes at each indicated time point after stimulation by H_2_O_2_ treatment (Appendix A), suggesting that CGA may induce the expression of Nrf2 at the post-transcriptional level but not at the transcriptional level. No significant change in HO-1 protein was detected (Figure 4a). Therefore, a 1.0 µM concentration of CGA was used for the following experiments. Next, after three days exposure to CGA or vehicle, HUVECs were stimulated with/without H_2_O_2_ for 1 h. Regarding Nrf2, the translocation into the nucleus was observed only in the CGA+/H_2_O_2_+ group at 1 h after stimulation with H_2_O_2_ (Figure 4b). This translocation significantly increased in the CGA-/H_2_O_2_+ group compared with the CGA+/H_2_O_2_+ group. Six hours after stimulation with H_2_O_2_, Nrf2 was detected only in the cytoplasm in the CGA+ groups irrespective of co-incubation with H_2_O_2_ (Figure 4b). Regarding HO-1, the mRNA level was determined by qPCR at the indicated time (Figure 4c). HO-1 expression increased significantly in the CGA+/H_2_O_2_− and CGA−/H_2_O_2_+ groups compared with the CGA−/H_2_O_2_− group at 0 h. HO-1 expression tended to increase more in the CGA+/H_2_O_2_+ group than in the CGA−/H_2_O_2_+ group. (*p* = 0.08380) Given these observations, CGA induced HO-1 expression irrespective of co-incubation with H_2_O_2_. Half an hour and one hour after H_2_O_2_ stimulation, HO-1 expression increased more in the CGA−/H_2_O_2_+ group than in the CGA+/H_2_O_2_+ group. CGA also increased the protein expression of HO-1 (Figure 4d), as well as mRNA expression. However, 3 h after H_2_O_2_ stimulation, the HO-1 protein level increased only in the CGA-/H_2_O_2_+ group compared with all other groups. Further, protein expression of p-Akt increased in the CGA−/H_2_O_2_+ group compared with all other groups (Figure 4e).

### 2.5. CGA Attenuates Senescence of Vascular Endothelial Cells through the Nrf2/HO-1 Pathway

To investigate whether CGA attenuates vascular EC senescence through the Nrf2/HO-1 pathway, a specific HO-1 inhibitor (zinc protoporphyrin IX, ZnPP) was administered to AngII-induced mice. Body weight and pulse rate were unaltered among all groups. SBP was notably higher in the AngII/saline group compared with the saline/saline group. (Table 2) The favorable effect of CGA on vascular senescence was canceled by ZnPP in vivo (Figure 5a).

To further elucidate whether the upregulation of HO-1 induced by CGA confers attenuation of senescence, the effect of ZnPP was also examined in vitro. ZnPP showed augmented cellular senescence irrespective of CGA (Figure 5b). Moreover, the protein expressions of Sirt1, eNOS, PAI-1, and p21 in aorta were examined among each group. Treatment with CGA increased Sirt1 and eNOS but reduced PAI-1 and p21. ZnPP treatment cancelled all these effects of CGA (Figure 5c). Taken together, these results clearly show that CGA attenuates vascular EC senescence through the Nrf2/HO-1 pathway.

## 3. Discussion

In this study, we found for the first time that CGA attenuated vascular senescence, in association with the increase of Sirt1 and eNOS and with the decrease of p-Akt, PAI-1, p53, and p21. Furthermore, in vivo and in vitro, we revealed that ZnPP canceled the anti-senescence effect of CGA. Thus, our study demonstrated that CGA inhibited endothelial senescence by regulating the Nrf2/HO-1 pathway.

AngII is a commonly used as an oxidative stress model in animal study. Chronic infusion of AngII induces oxidative-associated vascular senescence [39]. Since many reports on mice with senescence induced by AngII have been published, we used the AngII-induced model to examine vascular senescence in vivo. To enhance oxidative stress in vitro, we used H_2_O_2_ to treat HUVECs, leading to cellular senescence. Recent investigations have demonstrated that CGA activates Sirt1 in ECs [40]. In the present study, as expected, CGA increased Sirt1 and eNOS expression. These molecules are closely associated with vascular senescence; therefore, CGA has, at leaset in part, an anti-aging effect. Since eNOS upregulated by CGA is a target downstream of activated Akt/PKB, the expression of p-Akt was examined. Although it was assumed that the phosphorylation status of Akt showed a similar tendency of the expression of eNOS, the result was different than expected. Akt was activated with H_2_O_2_, and CGA canceled the activation of Akt. Activation of Akt not only led to the excessive production of ROS but also increased the expression of HO-1. The breakdown of the original redox homeostasis by high levels of intracellular oxidation or anti-oxidation is probably an important factor for senescence. Pretreatment of CGA may restore the original intracellular redox homeostasis by modulating AKT1 phosphorylation. Previously, a different kind of polyphenol, Mogroside V, was reported to attenuate the activation of Akt [41].

Similarly, resveratrol is a plant polyphenol that activates Sirt1 and eNOS [40,42,43]. Activation of the Nrf2 and Sirt1 signaling pathways by resveratrol ameliorated unfavorable cellular conditions, such as renal injury due to oxidative stress and mitochondrial dysfunction caused by aging [35]. Resveratrol retards age-related cognitive decline through Sirt1 [44]. Thus, resveratrol has an anti-aging effect. In this sense, polyphenols may be commonly equipped with an anti-aging effect. However, it is still unclear whether the upregulation of Sirt1 and eNOS by CGA is caused by the same mechanism as that used by resveratrol. Further studies are needed to elucidate the precise mechanism of how CGA regulates Sirt1 and eNOS expression. Basically, polyphenols exist in nature, are non-invasive, and seem to have minimal side effects on human beings, although the toxicity of a few polyphenols at high concentrations have been reported in vitro, as seen in the present study [45].

Nrf2 is a transcription factor responsible for the regulation of cellular redox balance and protective antioxidant and phase II detoxification responses in mammals [46,47]. We hypothesized that CGA may enhance the levels of phase II enzymes. In this study, we examined the mRNA expression of typical phase II enzymes, Superoxide dismutase 1(SOD1), Catalase (CAT), NAD(P)H dehydrogenase [quinone] 1 (NQO-1), Glutamate-cysteine ligase catalytic subunit (GCLC), and Glutamate-cysteine ligase modifier subunit (GCLM), by qPCR, but failed to find any significant changes in most of the mRNA levels other than HO-1. It is known that HO-1 has the highest capability to diminish oxidative stress among the genes induced by Nrf2 and plays an important role to prevent many diseases caused by oxidative stress, such as cardiovascular diseases [48,49]. Thus, we focused on the Nrf2/HO-1 pathway regarding the anti-aging effect of CGA on vascular senescence in this study. 

We further investigated the link between CGA and the Nrf2/HO-1 pathway. The in vitro study clearly demonstrated that treatment with CGA enhanced Nrf2 in the cytoplasm due to inhibition of its degradation by CGA. This result suggests that CGA makes these cells ready to bear oxidative stress. Therefore, after stimulation with H_2_O_2_, Nrf2 translocated into the nucleus faster in the CGA-treated cells, compared to those without CGA. Consequently, translocated Nrf2 could increase HO-1 mRNA and protein earlier. This increased HO-1 could promptly diminish oxidative stress, resulting in the attenuation of vascular senescence. The fact that a specific HO-1 inhibitor canceled the beneficial effect of CGA confirmed the impact of CGA on the regulation of HO-1 in anti-aging. Taken together, the beneficial effects of CGA on endothelial cell senescence were, at least partly, dependent on the Nrf2/HO-1 pathway. Previously, a different kind of polyphenol, fisetin, was reported to inhibit the degradation of Nrf2 [45]. It appears that this property might be commonly inherent to polyphenols.

Catechins from tea, anthocyanins from blueberries, and curcumin from curry are well-known polyphenols. Coffee contains many kinds of polyphenols, as much as red wine [25]. A recent cohort study revealed the cardioprotective effect of a low dosage of resveratrol (10 mg/day) in patients with stable coronary artery disease [50]. Red wine contains 1–75 mg of trans-resveratrol/L [51]. Therefore, to obtain the significant effect of resveratrol as above, it is necessary to drink 1.5 L or more of red wine. As mentioned earlier, individuals who habitually drink at least 3–4 cups of coffee a day have a reduced risk of mortality, heart disease, cerebrovascular disease, and respiratory disease [19]. A diet rich in polyphenols, at least CGA, may slow down aging, decrease the risk of age-related diseases, and improve quality of life.

In conclusion, our study demonstrated that CGA inhibited endothelial senescence both in vivo and in vitro by regulating the Nrf2/HO-1 pathway. CGA may be a new therapeutic target to attenuate endothelial senescence. CGA may help in developing suitable intervention therapies that decelerate the pace of aging as well as diminish or defer the prevalence of age-related diseases.

## 4. Materials and Methods 

### 4.1. Mice and Study Protocol 

C57/BL6 female mice aged 14 ± 3 months old were purchased from the Jackson Laboratory (Bar Harbor, Cat. No. 000664, ME, USA). All mice were maintained in a barrier facility, and ambient temperature ranged from 20 to 24 °C. Mice were fed a diet and water ad libitum. The mice were divided into five groups; saline/saline (*n* = 4), saline/CGA (*n* = 4), saline/AngII (*n* = 4), CGA low/AngII (*n* = 4), and CGA high/AngII (*n* = 4). Saline or AngII (1000 ng/kg/min, Bachem, Cat. No. H-1705-0100, Switzerland) was infused via Alzet mini-osmotic pumps (Alzet, Model 2002, Cupertino, CA, USA) for 14 days. Mini-osmic pumps were implanted subcutaneously on the right flank, as described previously [52,53]. In the low and high CGA (Sigma-Aldrich, Cat. No. C3878, St. Louis, MO, USA) groups, the mice were administered 20 or 40 mg/kg/day CGA via oral gavage for 14 days from the initial day of AngII infusion. In the other groups, the mice were given saline via oral gavage for 14 days from the initial day of AngII infusion. The mice were sacrificed 14 days after AngII infusion. The aorta was removed after systemic perfusion with phosphate-buffered saline for histological examination. The experimental protocol was approved by the Ethics Review Committees for Animal Experimentation of Okayama University Graduate School of Medicine, Dentistry, and Pharmaceutical Sciences (OKU-2018875, approved on 20 July 2018).

### 4.2. Blood Pressure Measurement

SBP and pulse rate were measured by sphygmomanometry using a tail cuff system (Visitech Systems, BP-2000, Napa Pl, NC, USA) following a published protocol [54]. Conscious mice were introduced into a small holder mounted on a thermostatically controlled warming plate and maintained at 37 °C during measurement.

### 4.3. SA-β-Gal Staining

A Cellular Senescence Assay Kit (CELL BIOLABS, INC. Cat. No.CBA-230, San Diego, CA, USA) was used throughout according to the company’s instructions.

### 4.4. Cell Culture and Treatment

HUVECs were purchased from Lonza (Lonza, Cat. No. C2519A 01127: multi donor, Basel, Switzerland) and were grown in endothelial growth medium (EGM™-2 Bullet Kit™ Medium: Lonza Cat. No.CC-3162, Basel, Switzerland). Population doubling levels were calculated as described previously [55], and all experiments were performed at population doubling levels of 7 to 9. HUVECs were grown in a 75 cm^2^ collagen-coated flask to 80% confluence for three days in the absence or presence of 0.5–5.0 μM CGA. The media were changed at 0 and 2 days (pre-treatment). Then, HUVECs were washed three times and stimulated for 1 h with 100 μM H_2_O_2_. After the stimulation, HUVECs were cultured with EGM-2 in the absence or presence of 0.5–5.0 μM CGA for an appropriate time.

### 4.5. Cellular Senescence

To evaluate cellular senescence, HUVECs were plated at 1.0 × 10^4^ cells/well into an 8-well chamber slide (Iwaki, Cat. no. 5732-008, Shizuoka, Japan) after pre-treatment with or without CGA and stimulated by H_2_O_2_, as described herein. Then, HUVECs were cultured with EGM-2 in the presence or absence of 0.5–5.0 μM CGA for an appropriate time. The morphological change of cells was observed by a phase-difference microscope. SA-β-gal activity was analyzed by staining as described above. Immunoperoxidase staining was performed to evaluate oxidative stress in HUVECs using 8-OHdG antibody (Japan Institute for the Control of Aging NIKKEN SEIL, Cat. No. clone N45.1, Shizuoka, Japan). Reactivity of the antibodies with tissue antigens was detected using AEC and ImmPACT AEC HRP Substrate (Vector Laboratories, SK-4200, Burlingame, CA, USA) as described previously [52,53].

### 4.6. Cell Viability

HUVECs were plated at 5.0 × 10^3^ cells/well into a 96-well plate (Corning, Cat. no. 354407, NY, USA) after pre-treatment with CGA and stimulated by H_2_O_2_, as described above. After the stimulation, HUVECs were cultured with EGM-2 containing these compounds for three days. The cell viability in the cell proliferation and cytotoxicity assays was determined by the Cell Counting Kit-8 (CCK-8) (Dojin, Cat. no. 347-07621, Kumamoto, Japan) according to the company’s instructions.

### 4.7. Western Blotting 

Whole cell proteins were extracted from HUVECs or aortic tissue using lysis buffer (Cell Signaling, Cat. No. 9803, Beverly, MA, USA). Nuclear and cytoplasmic proteins were extracted from HUVECs using a nuclear extract kit (ACTIVE MOTIF, Cat. No. 40010, Carlsbad, CA, USA) according to the manufacturer’s instructions. Each sample was loaded onto 10% SDS-PAGE and transferred to polyvinylidene fluoride membrane immunoblotted with primary antibodies (Sirt-1; Cell Signaling Cat. No. 9475S, eNOS; Cell Signaling Cat. No. 9572S, PAI-1; Cell Signaling Cat. No. 11907S, p53; Cell Signaling Cat. No. 9282S, p-Akt; Cell Signaling Cat. No. 4060S, Nrf2; abcam Cat. No. ab62352, HO-1; abcam Cat. No. ab13243, HDAC; Cell Signaling Cat. No. 34589, and β-actin; Sigma Cat. No. A5441). Membranes were then incubated with appropriate secondary antibodies, and immune complexes were visualized using chemiluminescence (Merck Millipore, Cat. no. WBLUF0100, Cat. no. WBLUC0100, Burlington, MA, USA) and were quantified using a General Electric Imager (GE Healthcare, LAS 4000 mini, Chicago, IL, USA) [52,53].

### 4.8. Real-Time Polymerase Chain Reaction

mRNAs were extracted from HUVECs using RNeasy Mini kits (Qiagen, Cat. No. 74104, Hilden, Germany). Reverse transcription was performed using an iScript cDNA synthesis kit (Bio Rad, Cat. No. 1708891, Hercules, CA, USA). PCR reactions were performed with an ABI Step One Real-Time PCR System (Applied Biosystems, Quant Studio3, Foster, CA, USA) using Fast SYBR Green Real-time PCR Mixture (Applied Biosystems, Cat. No. 4385612, Foster, CA, USA). [52] Primers for Nrf2, kelch-like ECH-associated protein 1 (keap1), HO-1, SOD1, CAT, NQO-1, GCLC, GCLM, and Glyceraldehyde-3-phosphate dehydrogenase (GAPDH) are available commercially (Takara Bio Inc., Kusatsu City, Japan). Each sample was normalized to values for GAPDH mRNA expression (ΔΔ*C*_t_ method).

### 4.9. Reagents

From stock solutions of ZnPP (Wako Pure Chemical Industries, Cat. No. 167-13651, Osaka, Japan), an HO-1 inhibitor was diluted with 0.25 M NaOH to a final concentration of 10 μg/μL. For administration to mice, the stock solutions were diluted with phosphate-buffered saline by 1% (*v*/*v*), 20 μg/mouse. As the control of in vitro and in vivo treatments, vehicle was used; for the in vitro experiment, 0.25 M NaOH was added to the culture medium (final pH of the culture medium was 7.6 ± 0.3); for in vivo treatment, 1% (*v*/*v*) 0.25 M NaOH was used (final pH of the injection solution was 8.3 ± 0.3) [56].

### 4.10. Treatment with ZnPP In Vivo

The mice were divided into four groups; saline/saline (*n* = 5), saline/AngII (*n* = 5), CGA/AngII (*n* = 5), and CGA/AngII/ZnPP (*n* = 5). Saline or AngII (1000 ng/kg/min) was infused via Alzet mini-osmotic pumps for 14 days. Mini-osmic pumps were implanted subcutaneously on the right flank, as described previously [52,53]. In the CGA group, the mice were administered 40 mg/kg/day CGA via oral gavage for 14 days from the initial day of AngII infusion. In other groups, mice were given saline via oral gavage for 14 days from the initial day of AngII infusion. ZnPP (20 μg/mouse) was injected into the peritoneal cavities every other day for 14 days from the initial day of AngII infusion. All mice were sacrificed 14 days after AngII infusion [56].

### 4.11. Treatment with ZnPP In Vitro

To evaluate cellular senescence, HUVECs were plated at 1.0 × 10^4^ cells/well onto an 8-well chamber slide after pre-treatment with CGA and stimulated by H_2_O_2_ with or without ZnPP at 0.5 μM, as described herein. Then, HUVECs were cultured with EGM-2 in the absence or presence of 1.0 μM CGA with or without ZnPP at 0.5 μM for 24 h. The morphological change of cells was evaluated by a phase-difference microscope. SA-β-gal activity was analyzed by staining as described previously.

### 4.12. Statistics

All statistical analyses were performed using Sigma Plot v14.0 (Systat Software Inc., San Jose, CA, USA). Data are presented as the mean ± standard deviation or standard error of the mean where appropriate. Statistical significance between multiple groups was assessed by one-way or two-way analysis of variance followed by the Holm–Sidak post hoc test or Student–Newman–Keuls post hoc test. A *p* value < 0.05 was considered statistically significant.

## Figures and Tables

**Figure 1 ijms-21-04527-f001:**
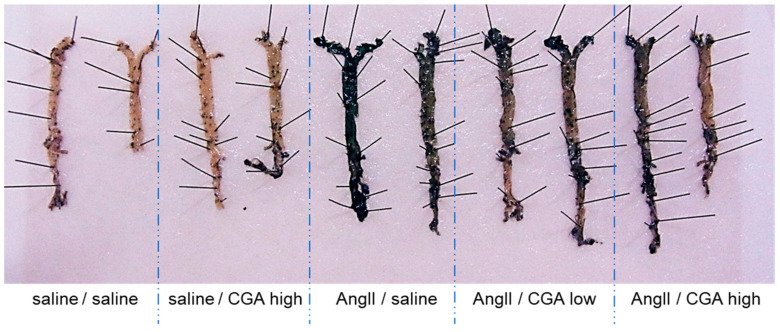
Effects of CGA on vascular senescence. SA-β-gal staining of aorta with or without CGA, low (20 mg/kg/day) or high (40  mg/kg/day), on 14 day after AngII infusion are shown.

**Figure 2 ijms-21-04527-f002:**
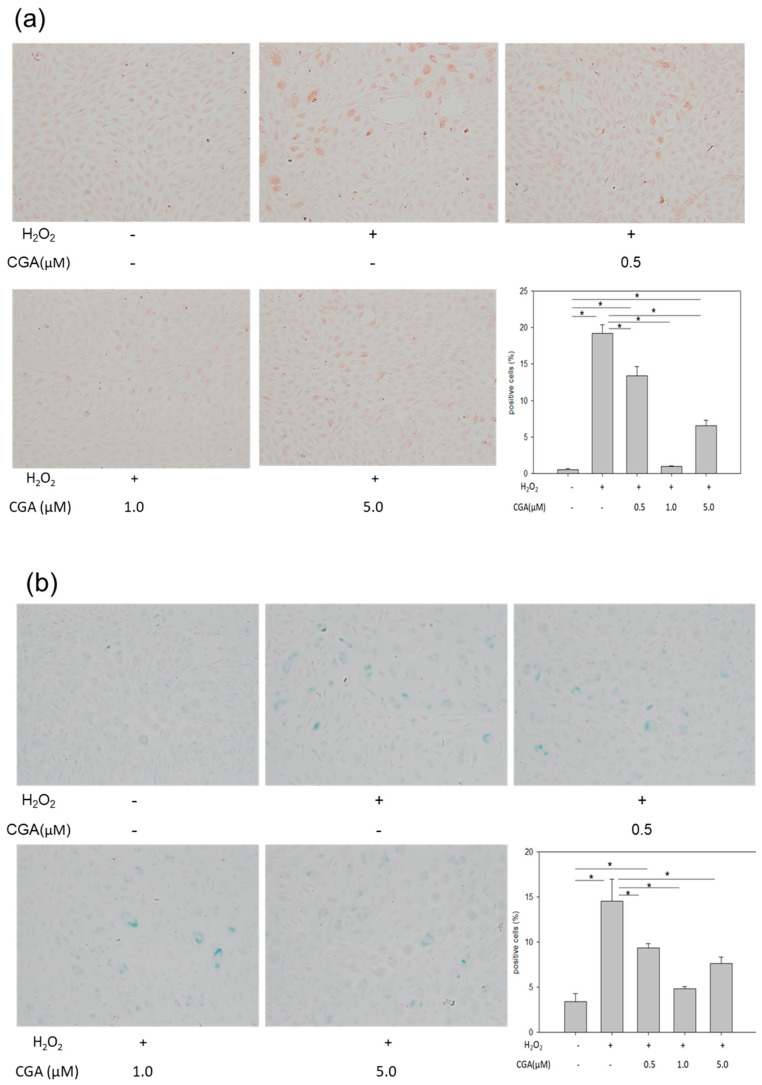
Effects of CGA on HUVECs. (**a**,**b**) Immunostaining of (**a**) 8-OHdG, (**b**) SA-β-gal staining, and (**a**,**b**) morphological changes Original magnification ×200. * *p* < 0.05 (*n* = 6) Each bar presents the mean ± SE of six experiments; (**c**) Cell proliferation was determined using a CCK-8 kit. * *p* < 0.05 vs. CGA-/H_2_O_2_-_,_
^#^
*p* < 0.05 (*n* = 3) Each bar presents the mean ± SE of three experiments; (**d**) Protein expression of Sirt1 and eNOS in CGA-treated HUVECs. * *p* < 0.05 (*n* = 3) Values represent the means ± SE of three experiments.

**Figure 3 ijms-21-04527-f003:**
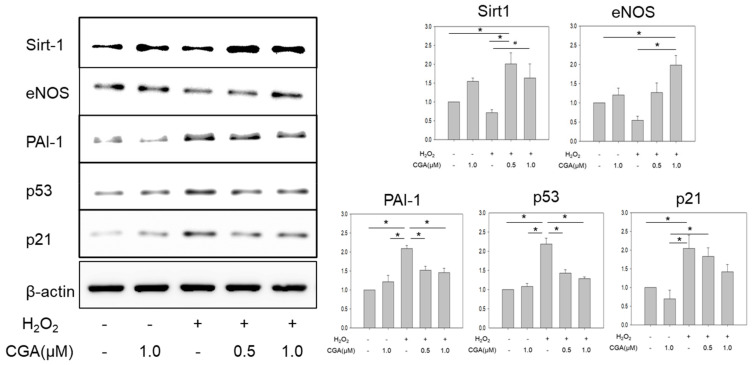
Effects of CGA on the senescence-related molecules. Protein expression of Sirt1, eNOS, PAI-1, p53, and p21, Quantitative analyses of the results. * *p* < 0.05_,_
^#^
*p* = 0.0618_,_ (*n* = 6). Values represent the means ± SE of six experiments.

**Figure 4 ijms-21-04527-f004:**
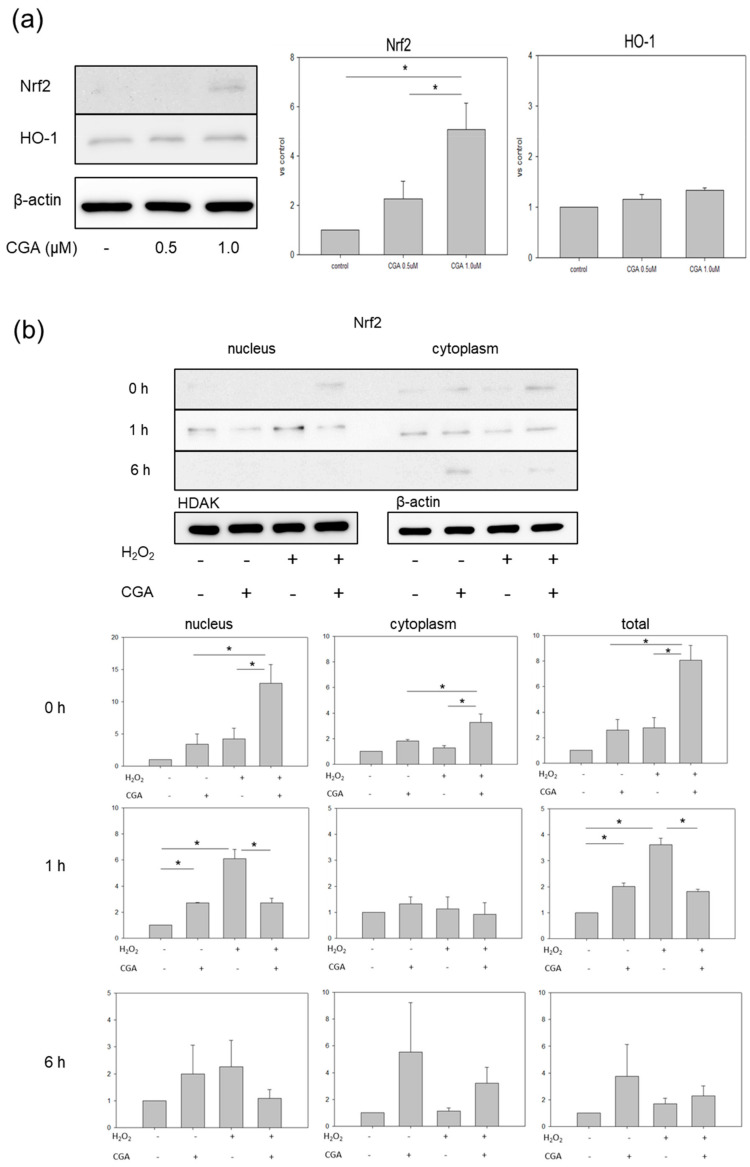
Effects of CGA on the expression of Nrf2-related proteins and mRNA. (**a**) Protein expression of Nrf2 and HO-1 in CGA-treated HUVECs (*n* = 3), * *p* < 0.05. Values represent the means ± SE of three experiments.; (**b**) Protein expression of Nrf2 in the nucleus and cytoplasm in CGA-treated HUVECs after stimulation with H_2_O_2_ (*n* = 3), * *p* < 0.05. Values represent the means ± SE of three experiments; (**c**) mRNA (*n* = 3), * *p* < 0.05. Each bar presents the mean ± SE of three experiments and (**d**) protein expression of HO-1 (*n* = 3). Values represent the mean ± SE of three experiments.; (**e**) Protein expression of p-Akt (*n* = 3). Values represent the means + SE of three experiments.

**Figure 5 ijms-21-04527-f005:**
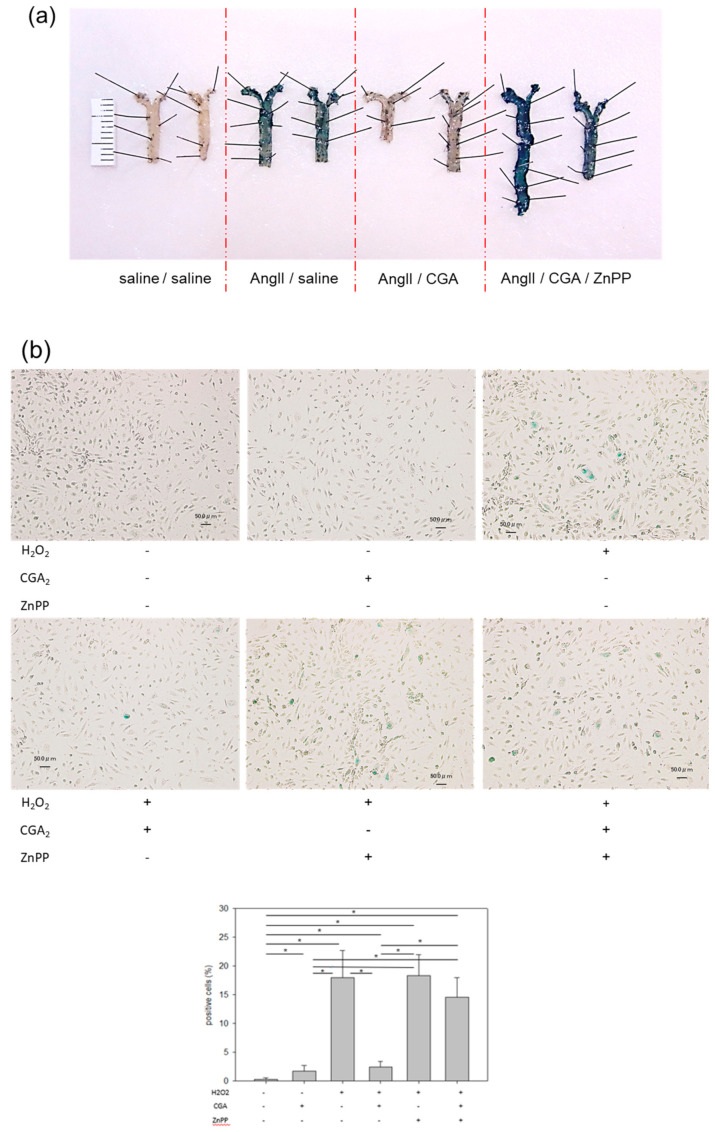
Effect of the HO-1 inhibitor on vascular senescence in vivo and in vitro. (**a**) SA-β-gal staining of aorta with or without CGA, and/or i.p. injection of ZnPP, and/or AngII for 14 days; (**b**) SA-β-gal staining of HUVECs co-incubated with or without H_2_O_2_, and/or CGA, and/or ZnPP. The scale bar indicates 50 µm. Each bar presents the mean ± SE of three experiments (*n* = 6) * *p* < 0.05; (**c**) Protein expression of Sirt1, eNOS, PAI-1, and p21 in aortas. (*n* = 3). * *p* < 0.05. Values represent the means ± SE of three experiments.

**Table 1 ijms-21-04527-t001:** Characteristics of the study mice.

	Saline/Saline	Saline/CGA High	Ang II/Saline	AngII/CGA Low	AngII/CGA High
N	4	4	4	4	4
BW (g)	33.5 ± 5.8	34.2 ± 5.1	29.9 ± 1.4	28.0 ± 1.4	31.6 ± 3.7
HR (bpm)	648 ± 28	662 ± 7	682 ± 43	660 ± 86	702 ± 23
SBP (mmHg)	104 ± 7	107 ± 9	129 ± 5 *^,#^	121 ± 13	119 ± 7

N: number, BW: body weight, HR: heart rate, SBP: systolic blood pressure, Data are the mean ± SD, * *p* < 0.05 vs saline/saline, ^#^
*p* < 0.05 vs saline/CGA high (one-way ANOVA on rank).

**Table 2 ijms-21-04527-t002:** Characteristics of the study mice.

	Saline/Saline	AngII/Saline	AngII/CGA	AngII/CGA/ZnPP
N	5	5	5	5
BW (g)	29.1 ± 5.5	28.7 ± 1.5	29.1 ± 3.8	27.2 ± 2.2
HR (bpm)	643 ± 62	663 ± 63	629 ± 99	564 ± 104
SBP (mmHg)	109 ± 9	127 ± 5 *	113 ± 9	112 ± 9

N: number, BW: body weight, HR: heart rate, SBP: systolic blood pressure, Data are the mean ± SD. * *p* < 0.05 vs saline/saline (one-way ANOVA on rank).

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
