# Peer review of "The Protective Effect of Chlorogenic Acid on Vascular Senescence via the Nrf2/HO-1 Pathway"

_ijms, 2020, doi:10.3390/ijms21124527_

Round 1
Reviewer 1 Report
The present study aimed to investigate possible anti-senescence effect of chlorogenic acid (CGA), a type of polyphenol contained coffee, using angiotensin II infusion mice model and H2O2-challenged HUVECs.
The present study demonstrated CGA inhibited senescence in both in vivo and in vitro models. Some underlying mechanisms were mainly investigated using HUVECs. The authors propose that the Nrf2/HO-1 pathway mediates anti-senescence effect of CGA. This conclusion is based mainly on the observations with a HO-1 inhibitor ZnPP. However, the effect of CGA on the expression of HO-1 is unclear according to the current investigations (Figure 4c and d). Figure 4c suggest that the inhibition of HO-1 expression is associated with the anti-senescence effect of CGA under H2O2 challenge. The western blot analysis does not readily support the notion that CGA up-regulated HO-1 expression. The critical drawback of the present study is the lack of convincing evidence that support the involvement of Nrf2/HO-1 in the anti-senescence effect of CGA.
Major points
1. The relevance of angiotensin II treatment in vivo to the H2O2 challenge in vitro remains unclear. Authors discussed the role of oxidative stress in the pro-senescence effect of angiotensin II. It is recommended to obtain some experimental evidence to support their relevance. It is recommended to examine the protective effect of anti-oxidants (N-acetyl-cysteine) or H2O2 scavenger (catalase) on the pro-senescence effect of angiotensin II in vivo. Alternatively, the pro-senescence effect of angiotensin II is recommended to be examined in HUVECs.
2. The western blot results (Figure 4a, b, d, e and Figure 5c) should be quantitatively evaluated. It is especially important in order to conclude the mechanistic involvement of Nrf2 and HO-1 in the anti-senescence effects of CGA. The notion that CGA up-regulates HO-1 expression is not plausibly supported by the data. The rationale for investigating the involvement of HO-1 in anti-senescence effect of CGA therefore remains unclear. Moreover, the notion that CGA facilitated the nuclear translocation of Nrf2 remains unconvincing. The quantitative evaluation of the western blot data is therefore critical for the conclusion of the present study.
3. Figure 4b also contains many uncertainty.
(1) It is unclear what time 0 h indicates. Is it before H2O2 challenge or just after application of H2O2? If it is the latter case, "0 h" may be misleading. Please specify when the data were exactly obtained, such as 1 min after H2O2 etc.
(2) It is unclear why both nuclear and cytoplasmic expressions of Nrf2 increase under the condition of H2O2(-)/CGA (-).
(3) The total level of Nrf2 should also be evaluated.
(4) The western blot data of nuclear and cytoplasmic expression of Nrf2 do not support the authors' conclusion that CGA accelerated the nuclear translocation of Nrf2 after H2O2 challenge. The quantitative evaluation of western blot data is therefore essential as mentioned above in the comment 1.
4. Figure 4c
The expression of HO-1 mRNA was suppressed by CGA treatment under the challenge with H2O2. This observation rather suggest that the inhibition of HO-1 by CGA is associated with anti-senescence effect of CGA. Therefore, the rationale for investigating the effect of the HO-1 inhibitor becomes unclear.
5. Figure 5
(1) The direct effect ZnPP is recommended to be examined in vivo and in vitro. If ZnPP per se induces senescence, then it is more probable that two events are independent. (2) The effect of CGA on the Nrf2 and HO-1 expression should be quantitatively evaluated as mentioned in the comment 2. The
Minor points
1. The figure legends should indicate in which way the data are presented, mean +/- SD or SEM. The experimental number should also be specified.
2. The statement that treatment with CGA suppressed their expression in dose-dependent manner (P. 4, L. 9) is inconsistent with Figure 3, especially for PAI-1 and p53. Please clarify this inconsistency.
3. All data in the supplemental figure should be mentioned in the main text. Currently, the observation with Nrf2 was only described.
4. The rationale for investigating the effect of CGA on the level of Akt phosphorylation is unclear. Please clarify this.
Reviewer 2 Report
The authors examine the effect of chlorogenic acid polyphenol (CGA) in vivo and in vitro in the context of vascular biology. Vascular protection effect associated to senescence of CGA is observed in response to stimulation of renin-angiotensin system with Angiotensin II peptide. Defects on relevant markers of cellular senescence are reverted by CGA in H2O2-induced senescent HUVEC cells. They then focus on Nrf2 and hemo oxygenase as a putative molecular mechanism for this effect and found that zinc protoporphyrin IX (ZnPP) Nrf2 inhibitor blocks the beneficial effect of CGA on blood pressure.
Fig 2a and 2b. Please provide quantification of positive cells and statistical analysis for each condition. The size of images might be increased for better visualization.
Figure 3. In the results section, the effect (including statistic analysis) of CGA on each marker should be explained in more detail.
Figure 4c. For better visualisation, treatment conditions might be provided on the x axis.
Figure 5b. Same as in Fig 2a and 2b.
Figure 5c. Quantification of signals should be provided. Quality of eNOS blot is not good enough for publication. This blot should be repeated or removed from the figure.
Author Response
Response to Reviewer #2’s Comments: ijms-831208
Title: The Protective Effect of Chlorogenic Acid on Vascular Senescence via Nrf2/HO-1 Pathway
We would like to express our deep appreciation to the Editor and Reviewers for their constructive comments to improve our manuscript. We have responded to the Reviewers’ questions and comments (italicized in the Response) point-by-point below, and modified the manuscript. Changes in the revised manuscript are highlighted in red. We hope this revised manuscript is now acceptable to the Reviewers.
Reviewer #2
The authors examine the effect of chlorogenic acid polyphenol (CGA) in vivo and in vitro in the context of vascular biology. Vascular protection effect associated to senescence of CGA is observed in response to stimulation of renin-angiotensin system with Angiotensin II peptide. Defects on relevant markers of cellular senescence are reverted by CGA in H2O2-induced senescent HUVEC cells. They then focus on Nrf2 and hemo oxygenase as a putative molecular mechanism for this effect and found that zinc protoporphyrin IX (ZnPP) Nrf2 inhibitor blocks the beneficial effect of CGA on blood pressure.
Thank the reviewer #2 so much for many constructive comments to improve our manuscript. These are the very idea what we would like to describe in this manuscript. We here made point-by-point answers.
Fig 2a and 2b. Please provide quantification of positive cells and statistical analysis for each condition. The size of images might be increased for better visualization.
Thank you so much for constructive comments. In accordance with your comments, we provided quantification data of positive cells and the result of the statistical analysis (Page 4, Figure 2(a) and (b)). We also changed the images. (Page 4, Figure 2(a) and (b)).
Figure 3. In the results section, the effect (including statistic analysis) of CGA on each marker should be explained in more detail.
Thank you for important comment. In order to confirm our data, we performed 3 more independent experiments, finally, data of a total 6 independent western blotting were obtained. First, we have provided quantitative analyses of these blots. (Figure 3b, Page 5) with a star mark when a significant difference presents and a hash tag when a tendency of significance. Second, we have modified the explanation in Figure 3 (Page 5, Line9) and figure legends of Figure 3 (Page 5, Line16-17).
Figure 4c. For better visualisation, treatment conditions might be provided on the x axis.
Thank you for valuable comment. In accordance with your comment, we changed the figure as you indicated, treatment conditions present in x axis. (Figure 4(c), Page 7)
Figure 5b. Same as in Fig 2a and 2b.
Thank you so much for constructive comments. In accordance with your comment, we provided quantification data of positive cells and the result of the statistical analysis (Figure 5(b) Page 9). We also changed the images. (Figure 5(b), Page 9).
Figure 5c. Quantification of signals should be provided. Quality of eNOS blot is not good enough for publication. This blot should be repeated or removed from the figure.
Thank you so much for constructive comments. After deeply consideration of the significance of eNOS data in this figure, we finally decided to remove the blot. (Figure 5c, Page9).
Round 2
Reviewer 1 Report
The manuscript has been satisfactorily revised.